# Activation of Persulfate for Degrading Tetracycline Using the Leaching Residues of the Lead-Zinc Flotation Tailing

**DOI:** 10.3390/polym14142959

**Published:** 2022-07-21

**Authors:** Jun Wang, Xiaocui Wen, Shaojun Jiang, Tao Chen

**Affiliations:** 1Fankou Lead-Zinc Mine, Shenzhen Zhongjin Lingnan Non-Ferrous Metal Company Limited, Shaoguan 512000, China; wangjun_198702@163.com; 2School of Environment, South China Normal University, University Town, Guangzhou 510006, China; xiaocui.wen@m.scnu.edu.cn (X.W.); shaojunj93@163.com (S.J.); 3Guangdong Provincial Key Laboratory of Chemical Pollution and Environmental Safety & MOE Key Laboratory of Theoretical Chemistry of Environment, South China Normal University, Guangzhou 510006, China

**Keywords:** leaching residues, tetracycline, persulfate activation, oxidative degradation, solid waste

## Abstract

Inappropriate disposal of leaching residues from the lead-zinc tailings recovery process may result in environmental pollution. Its recycling and reuse remain a prevalent topic in environmental science and technology. It was roasted to prepare leaching residues-based materials (TLRS) in this work, and the TLRS were creatively used as the catalyst to active sodium persulfate (PS) to degrade organic pollutants. Degradation of tetracycline using the TLRS–PS system was evaluated, and the treating parameters were optimized. Roasting resulted in the exposure of active sites on TLRS surface, in which transition metals can donate electrons to PS to form SO_4_^·−^. SO_4_^·−^ can further react with OH^−^ to form ·OH. Formation of these radicals was confirmed by both quenching experiments and EPR analysis. Under optimized conditions, 85% of the TC can be degraded in 3.0 h, and ~50% of degraded TC was mineralized to CO_2_ and H_2_O. The performance of TLRS barely changed after four reuses, suggesting the chemical stability of TLRS. The presence of dissolved substance in the water matrix could weaken the performance of the TLRS–PS system. A mechanism of TC degradation was proposed based on the experimental results and literature. These preliminary results provide us new insight on the reuse of lead-zinc flotation tailings.

## 1. Introduction

Antibiotics have played a highly prominent role in social civility and environmental health over the past century [1]. However, superabundant antibiotics, including tetracycline (TC), released into environmental matrices, have become a notable global environmental problem [2]. China uses up to 180,000 tons of antibiotics for both human and agricultural purposes. Therefore, it is critical to remove superabundant antibiotics such as TC from environmental media. To date, various treatment methods have been applied, including photodegradation [3], biodegradation [4], catalytic oxidation [5], and advanced oxidation processes (AOPs). With the free radicals (OH, SO_4_^·−^, O_2_^·−^, etc.) generated within the AOPs process, TC would be completely decomposed or mineralized. It was a most effective way to remove the TC among the methods reported in a specific environment. Free radicals can be generated with the activation of persulfate (PS) [6], and the PS can be activated by various methods, including UV–visible light, heat, bases, microwaves, carbon materials, and transition metal-containing materials [7,8]. However, these methods cannot be applied due to their high cost for the raw materials. Therefore, it is of great environmental significance to seek materials that allow for a multiple activation strategy and can be mass-produced for the heterogeneous activation of PS.

Tailing from lead-zinc mines is a serious environmental problem in many regions worldwide [7,9]. According to statistics, the amount of lead and zinc sulfide tailings generated in China is as high as 29 million tons, and the accumulated stockpile is over 1 billion tons [9]. Through the resource recovery process, sulfur, iron, and other valuable metals have been utilized, and the associated environmental risks were thoroughly eliminated [10,11,12,13]. While some residues were discarded, the discarded residue will be another pollution source without effective disposal [14]. In our previous study, the leaching residue was used as raw material to produce active silicon materials. The latter, a resourceful product from tailings, possesses highly available SiO_2_ content, rich porous structure, various forms of transition metals (Fe-, Mn-based metals, etc.), and their oxide sites, offering great potential for the immobilization of heavy metals [15,16,17]. Apart from this, Anbia et al. [18] reported that Fe_3_O_4_–copper-functionalized silicon nanowires from silicon powder were utilized as a catalyst for phenol degradation with an efficiency of approximately 100%. Highly doped silicon materials have also been shown to be capable of the post-catalytic degradation of methylene blue and anionic (methyl orange) dyes [19]. In addition, mine-based active Fenton-like materials have been already reported as capable of degrading TC and other antibiotics in water [20].

However, there are rare works focused on the use of tailings slag-based materials as PS activators for TC removal to date. Consequently, combining previous research on the leaching residue and the composition characteristics of active silicon materials, we conjectured that the composite obtained by the roasted leaching residue of the tailings would be an effective catalytic material as a PS activator to remove organic pollutants, which may simultaneously remove the target pollutants and obtain a potential amendment from the perspective of resource recycling.

In this study, the leaching residues of the lead-zinc (Pb-Zn) tailings were reused, they were roasted with the addition of an alkali material as an activator at some conditions. After the leaching residues-based materials (TLRS) were synthesized, they were characterized and used as the matrix for the decomposition of TC. During this study, the effect of TC concentration, pH value, and dosage of TLRS and persulfate on the degradation of TC were evaluated. In addition, the degradation mechanism of TC was evaluated by quenching experiments and EPR capture experiments. This research will strengthen the application of leaching residues in the TC decomposition process and provide a more effective way to degrade antibiotics as well.

## 2. Materials and Methods

### 2.1. Chemicals and Materials

Chemicals used were purchased from the market. Tetracycline hydrochloride (TC, AR, 99%), HCl (AR, 37%), NaOH (AR, 96%), tert-butyl alcohol (TBA), furfuryl alcohol (FFA), CH_3_OH and *p*-benzoquinone (BQ), and 5,5-dimethyl- L-pyrroline N-oxide (DMPO) were supplied by Aladdin (Shanghai, China). Sodium persulfate (PS, AR, 99%) and polyethylene glycol (PEG-4000) were obtained from the Damao Chemical Reagent Factory (Tianjin, China). The purities of these chemicals were all analytical grade or better. Ultrapure water generated from the Millipore system (Bedford, USA) was used for preparing the TC stock solutions and conducting the experiments.

Leaching residues of lead-zinc flotation tailings were collected from the lead-zinc tailings in the Fankou mine (Shaoguan, China). Residues were air-dried and milled to pass through 200 mesh nylon sieves. After that, the residues were modified by mixing the residues, Na_2_CO_3_, and CaCO_3_ at the mass ratio of 100:30:20 uniformly and roasting in a muffle furnace (PVSgr-20-2000, Shanghai Haoyue Electric Furnace Technology Co., Ltd. Shanghai, China) at 600 °C for 30 min. Then, the modified residues were cooled in air and washed with ultrapure water to adjust their pH to near neutral and remove the dissolved salts. After that, the mixture was dried and stored for further experiments. The mixture was named as leaching residues-based materials (TLRS).

### 2.2. Experimental Procedures

All experiments were conducted in 500 mL beakers at room temperature under mechanical stirring (150 rpm). TLRS were added to TC solutions at predetermined dosages. After equilibrium adsorption (24 h), PS was added to the mixtures to initiate the degradation process. Reaction mixtures were sampled from the reaction system at time intervals, filtered to pass through 0.22 μm polyether sulfone filter. The filtrate was collected and stored for the analysis of residual TC. Effects of initial pH, TLRS dosages, PC, and TC concentrations on the degradation of TC were evaluated by varying these parameters. The roles of active radicals on TC degradation were investigated by quenching experiments, where TBA, MeOH, FFA, and BQ were used as the scavengers [21]. The solid of the reaction mixture was collected, centrifuged, and air-dried for reusability experiments. All experiments were carried out in triplicate.

### 2.3. Analytical Methods

The TC concentrations were analyzed by Agilent 1260 high performance liquid chromatography (HPLC, Shimadzu, Kyoto, Japan) equipped with a diode array detector (DAD). The wavelength was set at 270 nm. TC in the reaction mixtures was separated on a Zorbax SB-C18 column (4.6 mm × 250 mm, 5 μm). A mixture of water (A) and 10 mM oxalic acid (B) was used as the mobile phase. The volume ratio of A and B was kept constant at 75:25. The changes of total organic carbon and fluorescence spectrum during the degradation of TC were measured by TOC-500 total organic carbon (TOC) analyzer (TOC-500, Shimadzu, Kyoto, Japan) and Hitachi F-4600 fluorescence spectrometer (Hitachi F-4600, Tokyo, Japan) [22,23], respectively. Active radicals generated in the system were in situ monitored by Bruker EMX-E electron paramagnetic resonance spectroscopy (EPR, Bruker EMXplus, Karlsruhe, Germany) [24]. The characteristics of the MLFTs before and after degradation of TC were characterized by Hitachi SU8020 field emission scanning electron microscopy (FE-SEM, Hitachi, Tokyo, Japan), Bruker VERTEX 70 Fourier transform infrared spectroscopy (FTIR, MPA II, Karlsruhe, Germany), X-ray powder diffraction (XRD, Ultima IV XRD, Tokyo, Japan), XRF-1800 X-ray fluorescence spectrometry (XRF, Shimadzu), and ESCALAB 250 X-ray photoelectron spectroscopy (XPS, Thermo Kalpha, Waltham, MA, USA). The Zeta potential and specific surface area of MLFTs were analyzed by Malvern Zetasizer Nano ZS90 (Malvern, UK) and Micromeritics ASAP-2020Plus Brunauer-Emmett-Teller (BET, ASAP 2020, Norcross, GA, USA) analyzer.

## 3. Results and Discussion

### 3.1. Characterization of TLRS

TLRS consists of Si, Fe, Ca, Al, K, Mg, and Ti at the mass ratios of 75.2%, 6.8%, 5.8%, 2.3%, 0.4%, 1.4%, 0.4%, and 0.3% (represents as oxides), respectively (Appendix A). Figure 1a shows the morphology of the TLRS. TLRS are porous with rough surfaces (Figure 1a). Some regular rod-like structures exist in the surface (Figure 1a). The specific surface area of TLRS determined by BET method is 37.06 m^2^/g (Figure 1b), which is related to rough surfaces and porousness. Generally, a higher specific surface area of TLRS was considered to be more favorable for absorbing pollution. Figure 1c shows the FTIR spectra of the TLRS. Characteristic peaks at 3200–3400 cm^−1^ were attributed to the vibration of –OH, and the peak at 550 cm^−1^ could be ascribed to the vibration of the Fe-O bonds in the crystalline lattice of iron oxides (Figure 1c) (Lei et al., 2018b). Moreover, the absorption bands in the region of 1430 and 880 cm^−1^ refer to carbonates. The intense absorption band is most likely related to asymmetric vibrations of the Si-O-Si bridge bonds as well as to asymmetric and symmetric vibrations of the Si-O end bonds. Calcite (CaCO_3_, PDF#99-0022), nepheline (NaAlSiO_4_, PDF#99-0083), goethite (FeOOH, PDF#99-0055), and microcline (KAlSi_3_O_8_, PDF#99-0078) were the main crystallite phases of TLRS, while it contained small amounts of almandine (Fe_3_Al_2_(SiO_4_)_3_, PDF#99-0003) and high-cristobalite (SiO_2_, PDF#99-0038) (Figure 1d), which was further confirmed by XPS analysis (Figure 1e). The zeta potential of TLRS was close to pH 6.2, which was higher than the raw materials.

### 3.2. Degradation of TC

Figure 2a shows the effects of initial pH on the degradation of TC. The degradation efficiency of TC decreased sharply with the increase of pH, which decreased from 97% to 58% as the solution pH increased from 2.0 to 12.0 (Figure 2a). In addition, 84% of the TC was removed at the end of 3.0 h reaction with an initial pH of 7.0 (Figure 2a). Although lowering the initial pH facilitated the degradation of TC, it also consumed external acid, which may result in the rise of treatment cost. Thus, the initial pH (around 7.0) was not adjusted in the subsequent experiments. The isoelectric point of TLRS is at ~6.2 (Figure 1f), it carries positive charges when the solution pH is lower than 6.2, while it takes negative charges at a solution pH higher than 6.2. Lowering the solution pH can increase the positive charges on TLRS surface to enhance the adsorption of S_2_O_8_^2−^, and thus to accelerate the generation of active radials [21]. Meanwhile, more active radials generated (e.g., ·OH and O_2_^·−^) in the system may be quenched by OH^−^ at a higher solution pH [22]. Moreover, OH^−^ may competitively occupy the active sites on TLRS surface to reduce the adsorption of TC and activation of PS [25,26,27]. The removal efficiencies of TC decreased with its initial concentrations at determined dosages of TLRS and PS (Figure 2b), which was expected given that more active radicals were required to achieve the equal degradation efficiency at higher initial TC concentrations according to the chemical equilibrium.

Notably, ~11% of TC was removed in the absence of TLRS, while the removal efficiency of TC increased to 89% in the presence of 0.4 g/L TLRS (Figure 2c). TLRS at higher loadings can provide larger surface area for the adsorption of TC and more active sites for activating PS to degrade TC [22]. The PS concentration had a pronounced effect on the degradation of TC (Figure 2d). The removal efficiency of TC increased with the PS concentration at the range of 0–5.0 mM, while further increasing the PS concentration inhibited the degradation of TC. A larger number of active radicals can be generated at higher dosages of PS (0–5.0 mM), while at much higher dosages, PS can also quench the radicals formed in the system (5.0–20.0 mM) (Figure 2d) [27,28]. Thus, excess PS inhibited the degradation of TC. The TOC was reduced by 48% at the end of 3.0 h reaction at an initial TC concentration of 20 mg/L and initial pH of 7.0 in the presence of 0.4 g/L TLRS and 5.0 mM PS, suggesting that about half of the degraded TC was mineralized to CO_2_ and H_2_O. Together, these results indicated that TLRS was effective for the activation of PS and radicals formed were robust for degrading TC.

Appendix A evaluates the effect water has on the degradation of TC. Cl^−^, NO_3_^−^, and SO_4_^2−^ barely affected the degradation of TC, consistent with the literature [29]. This can be explained by the excess of radicals generated in the system, and the addition of Cl^−^, NO_3_^−^, and SO_4_^2−^ at low concentrations resulted in negligible reduction of radical concentrations. However, H_2_PO_4_^−^ and HCO_3_^−^ significantly inhibited the degradation of TC. H_2_PO_4_^−^ and HCO_3_^−^ can decompose to release H^+^ to increase the solution pH [8,20]. Meanwhile, H_2_PO_4_^−^ can also interact with iron ions to reduce the effective concentration of active iron ions. HCO_3_^−^ can react with ·OH to form CO_3_^·–^ which is a weaker oxidant than ·OH [21]. HA can block the active sites on TLRS surface and quench the active radicals in the system, both effects inhibited the degradation of TC (Appendix A) [20].

### 3.3. Reaction Mechanism

The roles of active radicals on TC degradation were evaluated by quenching experiments. MeOH, TBA, FFA, and BQ were used as scavengers [30]. MeOH and TBA react with SO_4_^·−^ and ·OH at the rate constants of 1.6–7.7 × 10^7^ and 1.2–2.8 × 10^9^ M^−1^s^−1^, and 4.8–7.6 × 10^8^ and 4.0–9.1 × 10^5^ M^−1^s^−1^, respectively, while the second rate constants for the reactions between FFA and ^1^O_2_, and BQ and O_2_^·−^ were in the range of 0.9–1.0 × 10^9^ M^−1^ s^−1^ [30,31,32]. The single addition of MeOH, TBA, FAA, and BQ resulted in the reduction of the removal of TC, the removal efficiencies decreased from 85% to 16%, 31%, 63%, and 72%, respectively (Figure 3a), suggesting SO_4_^·−^ and ·OH could be the main radicals in the system, which was further confirmed by EPR analysis (Figure 3b) [20,33,34,35]. DMPO–SO_4_ and DMPO–OH were observed the EPR spectrum (Figure 3b), and the intensity of DMPO–OH was higher than DMPO–SO_4_, suggesting SO_4_^·−^ and ·OH were all involved in the degradation of TC and SO_4_^·−^ formed can transform to ·OH in the reaction system [36].

TLRS after the degradation of TC was further characterized by XPS analysis (Figure 4). The elemental composition on TLRS surface remained unchanged after the activation of PS, suggesting the chemical stability of TLRS (Figure 4a). Three peaks located at 284.6, 286.1, and 288.5 eV were observed on the C 1s spectrum, which could be assigned to the carbon in C–C, C=O, and O-C=O states, respectively. C=O functional groups are electron-gaining functional groups, which can act as an intermediate for electron transfer to form free radicals (i.e., SO_4_^·−^ and ·OH) to promote the decomposition of TC [21,24]. Characteristic peaks on O1s spectrum at the binding energies of 530.8, 531.3, and 532.7 eV were attributed to the oxygen in surface hydroxyl, C-O, and C=O, respectively [37]. The disappearance of the peak at 530.8 eV after the degradation of TC suggested the release of hydroxyl. Peaks at the binding energies of ~710, ~718, and ~724 eV suggested the co-existence of Fe(II), Fe^0^, and Fe(III) in TLRS (Figure 4d) [8,38]. The reduced peak area at ~724 eV and increased peak area at ~710 eV at the end of the reaction indicated the transformation of iron species during the degradation process. The electron transfers within the iron species resulted in the activation of PS [20,24,38].

Taken together, mechanism for the degradation of TC was proposed based on our experimental results and literature (Figure 5) [24,26,33,37]. For phase 1, the transition metals and iron oxide in TLRS catalyzed the decomposition of PS to generate SO_4_^·−^ through electron transfer (S_2_O_8_^2−^ + Fe^0^→Fe^2+^ + 2SO_4_^·−^ + 2SO_4_^2−^, S_2_O_8_^2−^ + Fe^2+^→Fe^3+^ + SO_4_^·−^ + SO_4_^2−^, S_2_O_8_^2−^ + HSO_5_^−^ + M^n+^→M^(n−1)+^ + SO_4_^·−^ + SO_4_^2−^) (Figure 5). The electrons transfer from OH^−^ to SO_4_^·−^ resulted in the formation of ·OH (SO_4_^·−^ + OH^−^→·OH + SO_4_^2−^). ·OH and SO_4_^·−^ are strong oxidants, which can result in the oxidative degradation of TC (Figure 5), which can be directly reflected on the three-dimensional fluorescence spectrum [38,39]. In phase 2, regeneration of the active sites of TLRS made the activation cycle continue until PS was completely consumed. The yielded radical dot SO_4_^−^ attacked TC, and various intermediate products were generated, which made TC ultimately mineralize into CO_2_ and H_2_O [40]. The raw TC solution had two characteristic fluorescence peak areas at Em/Ex = 430 nm/330 nm and Em/Ex = 430 nm/230 nm, and the intensities for these two peaks were 1100 and 700, respectively (Figure 6). These two peaks disappeared at the end of 3.0 h reaction, confirming the degradation of TC [22,36,41], consistent with the TOC results.

### 3.4. Reusability Tests and Application in Real Water

Reusability is an important factor in determining the application of TLRS. The degradation efficiency of TC barely changed in the first three cycles, while it slightly decreased in the fourth cycle (Figure 7a). The reduced performance of TLRS could be attributed to the leaching of iron and transition metals (Appendix A) [42]. Meanwhile, the accumulation of degradation products can also occupy the active sites on the TLRS surface to reduce the degradation of TC [42]. The above results confirm that TLRS is structurally stable and reusable for the activation of PS and degradation of TC.

Two types of water matrix (i.e., tap water and Zhujiang river water) were used to evaluate the performance of the TLRS–PS system on degrading TC. Comparable TC removal efficiencies were achieved in tap water with that of in Mill-Q water (85% vs. 83%), while a slight decrease in the Zhujiang river water sample was observed (78%) (Figure 7b). Zhujiang river water contained much higher metal ions, anions, and dissolved organic matter than tap water and pure water (Appendix A), these dissolved substances can competitively occupy the active sites on TLRS surface to inhibit the generation of active radicals (mainly ·OH and SO_4_^·−^) and they can also scavenge the radicals formed [24,41]. The dosages of TLRS and PS can be flexibly adjusted to counteract the adverse effects brought by dissolved substances in the water matrix.

## 4. Conclusions

Leaching residues of lead-zinc flotation tailing were modified by roasting to active PS in the study. Degradation of TC using TLRS–PS was evaluated under various conditions. More active sites can be exposed on the TLRS surface through roasting to active PS. The electrons transfer between iron species, transition metals, and PS resulted in the generation of SO_4_^·−^, which can be further converted to ·OH. Approximately half of the degraded TC can be mineralized to CO_2_ and H_2_O, suggesting the robust oxidation ability of SO_4_^·−^ and ·OH. The performance of TLRS–PS barely degraded in the removal of TC after three reuses of TLRS. The presence of dissolved substances in water can weaken the performance of the TLRS–PS system through occupying the active site to reduce the generation of radicals and scavenging the radicals formed. This work shows us a new direction in the recycling of lead-zinc flotation tailing.

## Figures and Tables

**Figure 1 polymers-14-02959-f001:**
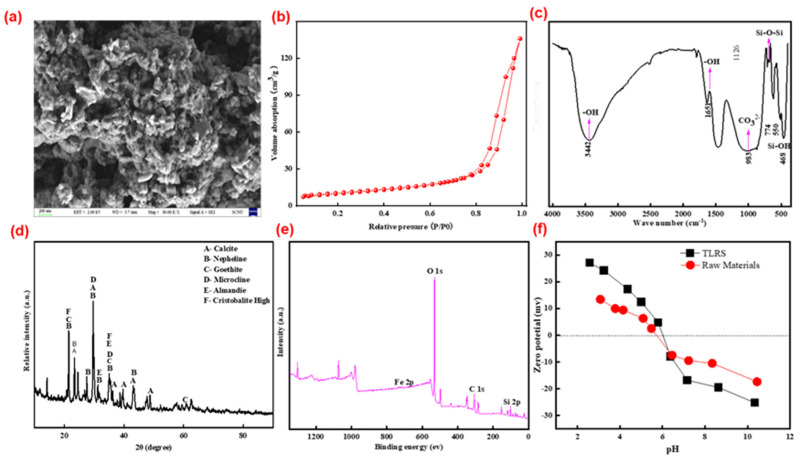
Characterization of TLRS: (**a**) SEM morphology, (**b**) N_2_ adsorption curve of MLTFS, (**c**) FTIR spectrum, (**d**) XRD pattern, (**e**) XPS spectrum, and (**f**) zeta potential of MLTFS.

**Figure 2 polymers-14-02959-f002:**
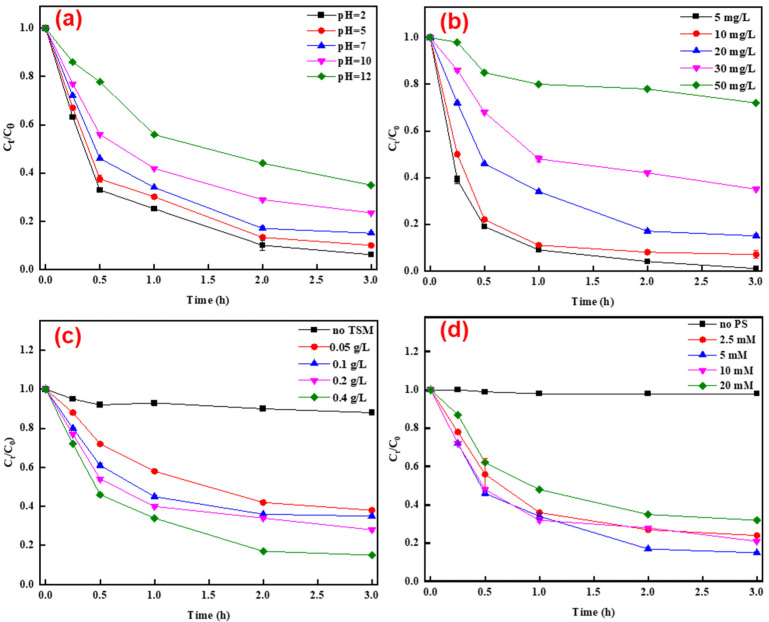
Degradation of TC under various conditions: (**a**) degradation of TC under different initial pH (PS = 5 mM, TC = 20 mg/L, TLRS = 0.4 g/L); (**b**) degradation of TC under various initial concentrations (initial pH of 7.0, PS = 5 mM, TLRS = 0.4 g/L); (**c**) degradation of TC under different TLRS loadings (initial pH of 7.0, PS = 5 mM, TC = 20 mg/L); and (**d**) degradation of TC under different PS loadings (initial pH of 7.0, TC = 20 mg/L, pH = 7, TLRS = 0.4 g/L).

**Figure 3 polymers-14-02959-f003:**
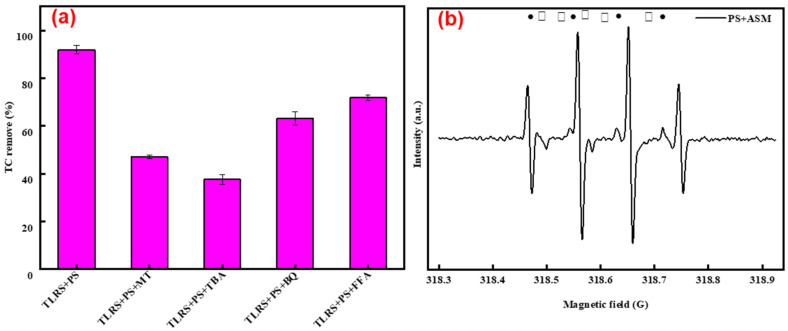
Identification of the active radicals using quenching experiments and EPR analysis: (**a**) degradation of TC in the presence of MeOH, TBA, FFA, and BQ (Experimental conditions: TC = 20 mg/L, PS = 5 mM, TLRS = 0.4 g/L, MeOH = TBA = BQ = FFA = 20 mM, and initial pH = 7); (**b**) EPR spectra generated in the system using DMPO as the trapping agent. (Experimental conditions: PS = 5 mM, TLRS = 0.4 g/L; black square: DMPO-OH; gray circle: DMPO-SO4.).

**Figure 4 polymers-14-02959-f004:**
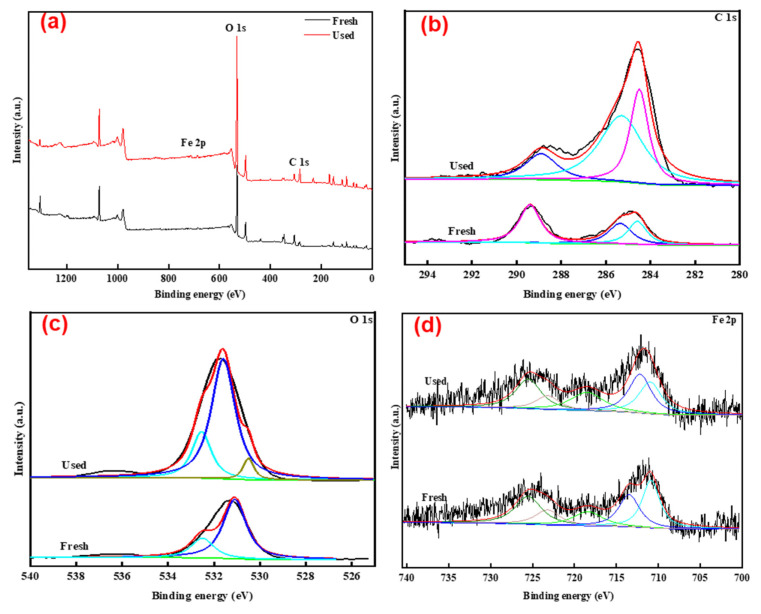
XPS spectra of the TLRS before and after the degradation of TC: (**a**) survey spectra; (**b**) C 1s spectrum; (**c**) O 1s spectrum; and (**d**) Fe 2p spectrum.

**Figure 5 polymers-14-02959-f005:**
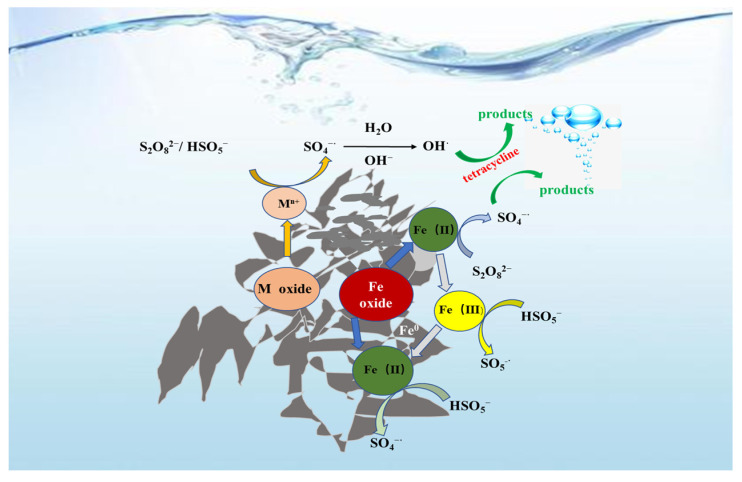
Proposed mechanism of the TC degradation in the TLRS/PS system.

**Figure 6 polymers-14-02959-f006:**
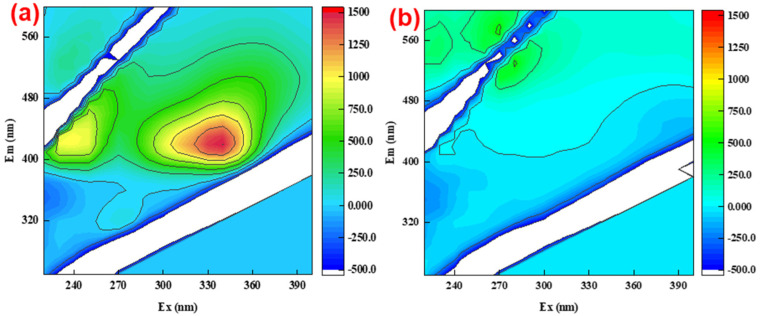
Three-dimensional fluorescence spectrum of TLRS/PS system before (**a**) and after (**b**) degradation of TC.

**Figure 7 polymers-14-02959-f007:**
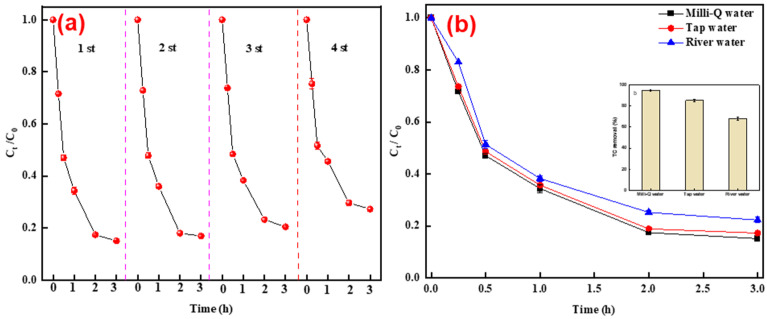
Degradation of TC in TLRS/PS using the reused TLRS (**a**) and in tap water and Zhujiang river water matrix (**b**) (respectively). Experimental conditions: TC = 20 mg/L, PS = 5 mM, MLTFS = 0.4 g/L, and initial pH of 7.

## Data Availability

Not applicable.

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
