# Peer review of "Activation of Persulfate for Degrading Tetracycline Using the Leaching Residues of the Lead-Zinc Flotation Tailing"

_polymers, 2022, doi:10.3390/polym14142959_

Round 1
Reviewer 1 Report
The subject of the paper is very interesting but needs minor revision.
1. Introduction: Include the quantitative information also.
2. Rewrite the keywords.
3. Include the purity of the chemicals used.
4. In figure 1(c): the y-axis has no unit; check it carefully.
5. In figure 1(d): Some of the XRD peaks not indexing. Check it carefully.
6. Check the caption of figure 1(b).
7. The authors should re-check the XPS results of the prepared samples.
8. In figure 4: Check the x-axis unit carefully.
9. The authors should improve the proposed mechanism using the recent references.
10. More typo errors in the manuscript. Check it carefully.
11. Check reference 40 carefully.
Author Response
Dear Editors and Reviewers:
Thank you for your letter and for the reviewers’ comments concerning our manuscript entitled “Activation of Persulfate for Degrading Tetracycline using the Leaching Residues of the Lead-zinc Flotation Tailing” (ID: polymers-1809703). These comments are all valuable and very helpful for revising and improving our paper, as well as the important guiding significance of our research. We have studied these comments carefully and have made corrections and we hope to meet with your approval. Revised portions are marked in red in the paper. The main corrections in the paper and the response to the reviewer’s comments are as following:
Reviewer#1
The subject of the paper is very interesting but needs minor revision.
- Introduction: Include the quantitative information also.
Response: In the introduction, we also have added the quantitative information of the antibiotics and the amount of accumulated tailings in the revised manuscript, that is “Antibiotics have played a highly prominent role in social civility and environment health over the past century [1]. However, superabundant antibiotics, including tetracycline (TC), released into environmental matrices, become a global environmental notorious problem[2]. China uses up to 180,000 tons of antibiotics for both human and agricultural purposes.…”and “Tailing from the lead-zinc mine is a serious environmental problem in many regions worldwide[9-10]. According to statistics, the amount of lead and zinc sulfide tailings generated in China is as high as 29 million tons, and the accumulated stockpile is over 1 billion tons [9]. …”
- Rewrite the keywords.
Response: we have rewritten the keywords that is “Leaching residues; tetracycline, persulfate activation; oxidative degradation; solid waste”.
- Include the purity of the chemicals used.
Response:We have added the the purity of the some chemicals in the 2.1 section. That is: “Chemicals used were purchased from the market. Tetracycline hydrochloride (TC, AR, 99%), HCl (ACS, 37%), NaOH (AR, 96%), tert-butyl alcohol (TBA), furfuryl alcohol (FFA), CH3OH and p-benzoquinone (BQ), 5,5-dimethyl- L-pyrroline N-oxide (DMPO) were supplied by Aladdin (Shanghai, China). Sodium persulfate (PS, AR, 99%) and polyethylene glycol (PEG-4000) were obtained from the Damao Chemical Reagent Factory (Tianjin, China). The purities of these chemicals are all in analytical grade or better. Ultrapure water generated from the Millipore system (Bedford, USA) were used for preparing the TC stock solutions and conducting the experiments.”
- In figure 1(c): the y-axis has no unit; check it carefully.
Response: We have deleted the unit of y-axis.
- In figure 1(d): Some of the XRD peaks not indexing. Check it carefully.
Response: We have added the indexing of the XRD peaks of Fig 1(d). Six substances were all shown in Fig.1d, including Calcite (CaCO3, PDF#99-0022), nepheline (NaAlSiO4, PDF#99-0083), goethite (FeOOH, PDF#99-0055), microcline (KAlSi3O8, PDF#99-0078) (Fe3Al2(SiO4)3, PDF#99-0003) and high-cristobalite (SiO2, PDF#99-0038).
Fig.1 d
- Check the caption of figure 1(b).
Response: We have changed the caption of Fig.1(b).
- The authors should re-check the XPS results of the prepared samples.
Response: Thanks for the reviewer’ reminder, We have carefuly checked the XPS results of the prepared samples and have confirmed it is correct.
- In figure 4: Check the x-axis unit carefully.
Response: Thanks for the reviewer’ reminder, We have changed the the x-axis unit in the revised manuscript.
- The authors should improve the proposed mechanism using the recent
Response: Thanks for the reviewer’ suggestion, we have carefully read recent references, which gave us much valuable information in improving our manuscript and were listed in the References in the revised manuscript. According to the reviewer’ suggestion, we have improved the proposed mechanism using the recent references to make the mechanism of this study highlight. Thus, the whole mechanism section has been rewritten as the following: “Taken together, mechanism for the degradation of TC was proposed based on our experimental results and literature (Fig. 5a)[25,27,34,38]. For phase 1, the transition metals and iron oxide in TLRS catalyzed the decomposition of PS to generate SO4•− through electron transfer (S2O82− + Fe0 → Fe2+ + 2SO4•− + 2SO42−, S2O82− + Fe2+ → Fe3+ + SO4•− + SO42−, S2O82− + HSO5− + Mn+ → M(n-1)+ + SO4•− + SO42−) (Fig. 5). The electrons transfer from OH− to SO4•− resulted in the formation of •OH (SO4•− + OH− → •OH + SO42−). •OH and SO4•− are strong oxidants, which can result in the oxidative degradation of TC (Fig. 5), which can be directly reflected on three-dimensional fluorescence spectrum [39,40]. In phase 2: regeneration of the active sites of TLRS made the activation cycle continue until PS was completely consumed. The yielded radical dot SO4·-attacked TC, and various intermediate products were generated, which made TC ultimately mineralize into CO2 and H2O [41].”
References:
Shi, X., Wang, L., Zuh, A. A., Jia, Y., Ding, F., Cheng, H., & Wang, Q. (2022). Photo-Fenton reaction for the degradation of tetracycline hydrochloride using a FeWO4/BiOCl nanocomposite. Journal of Alloys and Compounds, 903, 163889.
Sun, R., Yang, J., Huang, R., & Wang, C. (2022). Controlled carbonization of microplastics loaded nano zero-valent iron for catalytic degradation of tetracycline. Chemosphere, 135123.
Wang, T., Xue, L., Liu, Y., Fang, T., Zhang, L., & Xing, B. (2022). Insight into the significant contribution of intrinsic defects of carbon-based materials for the efficient removal of tetracycline antibiotics. Chemical Engineering Journal, 435, 134822.
- More typo errors in the manuscript. Check it carefully.
Response: Thanks for the reviewer’ reminder, we have changed the typo errors and marked in red in the revised manuscript.
- Check reference 40 carefully.
Response: We have added the number in the revised manuscript.

Reviewer 2 Report
The presented work is devoted to solving a complex of environmental problems: recycling of lead-zinc flotation tailings and decomposition of tetracycline using the material obtained from the waste.
Some questions and comments on the work:
1. In order to assess the potential of use, it is desirable to specify the amount of accumulated tailings? Are there tailings of similar composition in China? It would be interesting to show how universal the research conducted is or whether it has more of a regional aspect.
2. Not each instrument has a country-manufacturer or city. Need to give this information in a more single type.
3. Zoom in on the SEM image (Fig. 1 a).
4. If the authors give an isotherm of nitrogen adsorption (Fig. 1 b), then a more detailed description of the relationship should be given, since there are no comments in the text.
5. As far as the reviewer understands from the study, is the diffractogram in Fig. 1 f given a diffractogram of wastes after their heating at 600°C with calcium and sodium carbonates? A diffractogram of the original waste (before annealing) is desirable.
6. The IR spectrum shown in Fig. 1c is not correctly described. The absorption bands in the region of 1430 and 880 cm-1 refer to carbonates. The intense absorption band is most likely related to asymmetric vibrations of the Si-O-Si bridge bonds as well as to asymmetric and symmetric vibrations of the Si-O end bonds.
The results obtained in the presented work are of scientific and applied interest. The paper can be accepted for publication.
Author Response
Review#2:
The presented work is devoted to solving a complex of environmental problems: recycling of lead-zinc flotation tailings and decomposition of tetracycline using the material obtained from the waste.
Some questions and comments on the work:
- In order to assess the potential of use, it is desirable to specify the amount of accumulated tailings? Are there tailings of similar composition in China? It would be interesting to show how universal the research conducted is or whether it has more of a regional aspect.
Response: In the introduction, we also have added the information of the amount of accumulated tailings and the tailings of similar composition in China in the revised manuscript, that is “Tailing from the lead-zinc mine is a serious environmental problem in many regions worldwide [9-10]. According to statistics, the amount of lead and zinc sulfide tailings generated in China is as high as 29 million tons, and the accumulated stockpile is over 1 billion tons [9].”
In addition, Lead-zinc sulfide deposits contain a large amount of valuable metal elements, which are mainly found in sphalerite [(Zn,Fe)S], galena [PbS], chalcopyrite [CuFeS2] and nickel pyrite [(Fe,Ni)9S8]. During the development of lead-zinc sulfide ore, these heavy metal elements usually enter the tailings pond as trace impurity elements in the tailings in the form of co-association, which has laid a potential problem of heavy metal pollution in the tailings pond. Therefore, there is a need for resource utilization of these tailings.
- Not each instrument has a country-manufacturer or city. Need to give this information in a more single type.
Response: We have added a country-manufacturer or city for the each instrument in the revised manuscript.
The TC concentrations were analyzed by Agilent 1260 high performance liquid chromatography (HPLC, Shimadzu, Japan) equipped with a diode array detector (DAD). The wavelength was set at 270 nm. TC in the reaction mixtures was separated on a Zorbax SB-C18 column (4.6×250 mm, 5 μm). A mixture of water (A) and 10 mM oxalic acid (B) was used as the mobile phase. The volume ratio of A and B was kept constant at 75: 25. The changes of total organic carbon and fluorescence spectrum during the degradation of TC was measured by TOC-500 total organic carbon (TOC) analyzer (TOC-500, Shimadzu, Japan) and Hitachi F-4600 fluorescence spectrometer (Hitachi F-4600, Tokyo, Japan)[23,24], respectively. Active radicals generated in the system were in-situ monitored by Bruker EMX-E electron paramagnetic resonance spectroscopy (EPR, Bruker EMXplus, Germany)[25]. The characteristics of the MLFTs before and after degradation of TC were characterized by Hitachi SU8020 field emission scanning electron microscopy (FE-SEM, Hitachi, Japan), Bruker VERTEX 70 Fourier transform infrared spectroscopy (FTIR, MPA II, Germany), X-ray powder diffraction (XRD, Ultima IV XRD, Japan), XRF-1800 X-ray fluorescence spectrometry (XRF, Shimadzu), and ESCALAB 250 X-ray photoelectron spectroscopy (XPS, Thermo Kalpha, USA). The Zeta potential and specific surface area of MLFTs were analyzed by Malvern Zetasizer Nano ZS90 (England) and Micromeritics ASAP-2020Plus Brunauer-Emmett-Teller (BET, ASAP 2020, USA) analyzer.
- Zoom in on the SEM image (Fig. 1 a).
Response: We have zooed the SEM image (Fig.1a).
- If the authors give an isotherm of nitrogen adsorption (Fig. 1 b), then a more detailed description of the relationship should be given, since there are no comments in the text.
Response: we have analyzed the combined study detailed description of the nitrogen adsorption in the revised manuscript. that is “The specific surface area of TLRS determined by BET method is 37.06 m2/g (Fig. 1b), which related to rough surfaces and porous. Generally, higher specific surface area of TLRS was considered to be more favorable for absorbing pollution.”
- As far as the reviewer understands from the study, is the diffractogram in Fig. 1 f given a diffractogram of wastes after their heating at 600°C with calcium and sodium carbonates? A diffractogram of the original waste (before annealing) is desirable.
Response: Thanks for the reviewer’ valuable suggestion, we carefully consider your suggestions and combine them with this study. We have added the diffractogram of the original waste of the Fig. 1 f. The corresponding modifications are shown below:
- The IR spectrum shown in Fig. 1c is not correctly described. The absorption bands in the region of 1430 and 880 cm-1 refer to carbonates. The intense absorption band is most likely related to asymmetric vibrations of the Si-O-Si bridge bonds as well as to asymmetric and symmetric vibrations of the Si-O end bonds.
Response: Thanks for the reviewer’ reminder, We have changed this error described in the revised manuscript that is “Figure 1c shows the FT-IR spectra of the TLRS. Characteristic peaks at 3200–3400 cm−1 were attributed to the vibration of –OH, and the peak at 550 cm−1 could be ascribed to the vibration of the Fe-O bonds in the crystalline lattice of iron oxides (Fig. 1c) (Lei et al., 2018b). Moreover, the absorption bands in the region of 1430 and 880 cm-1 refer to carbonates. The intense absorption band is most likely related to asymmetric vibrations of the Si-O-Si bridge bonds as well as to asymmetric and symmetric vibrations of the Si-O end bonds.”
The results obtained in the presented work are of scientific and applied interest. The paper can be accepted for publication.
